# HBcrAg Predicts Hepatocellular Carcinoma Development in Chronic B Hepatitis Related Liver Cirrhosis Patients Undergoing Long-Term Effective Anti-Viral

**DOI:** 10.3390/v14122671

**Published:** 2022-11-29

**Authors:** Kuo-Chin Chang, Ming-Tsung Lin, Jing-Houng Wang, Chao-Hung Hung, Chien-Hung Chen, Sherry Yueh-Hsia Chiu, Tsung-Hui Hu

**Affiliations:** 1Division of Hepato-Gastroenterology, Department of Internal Medicine, Kaohsiung Chang Gung Memorial Hospital and Chang Gung University College of Medicine, Kaohsiung 833, Taiwan; 2Department of Health Care Management, College of Management, and Healthy Aging Research Center, Chang Gung University, Taoyuan 333, Taiwan

**Keywords:** chronic B hepatitis, hepatitis B core-related antigen, hepatocellular carcinoma, liver cirrhosis, anti-viral therapy

## Abstract

Hepatitis B core-related antigen (HBcrAg) is a predictor of hepatocellular carcinoma (HCC) in chronic hepatitis B (CHB) patients. Studies on anti-viral therapy have shown that the use of NUC therapy in HBV patients could reduce the incidence of HCC. However, the incidence of HCC continues to increase after long-term anti-viral therapy. The relationship between HBcrAg and HCC development in CHB-related liver cirrhosis (LC) patients undergoing long-term anti-viral therapy is still unclear. This study enrolled 1108 treatment-naïve CHB patients diagnosed with HBV-related LC receiving NUC therapy from April 1999 to February 2015. The baseline biomarkers, disease history, and following results were collected by the hospital. Among the 1108 patients, 219 developed HCC within a median follow-up period of 6.85 years. A multivariable Cox regression model was used, with adjustment for age, gender, FIB-4, DM, and HBsAg-HQ. The adjusted hazard ratios for the HBcrAg tertile levels were 1.70 (95%CI: 1.21, 2.39) and 2.14 (95%CI: 1.50, 3.05) for levels 3.4–4.9 and >4.9 logU/mL, respectively, compared with levels ≤3.4. The effect of the HBcrAg level on HCC incidence was found to be significantly modified by HBsAg-HQ, where lower HBsAg-HQ (≤ 3) values were associated with a significantly higher risk, but HBsAg-HQ levels >3 were not. Our results highlight that, after adjustment for potential confounding factors, patients with CHB-related LC and higher HBcrAg levels are at significant risk for HCC development, even while undergoing long-term effective anti-viral therapy. The HBcrAg level is therefore an independent risk factor for HCC development, especially for patients with HBsAg-HQ levels <3.

## 1. Introduction

Chronic B infection (CHB) remains a major public health problem worldwide [1,2,3]. Approximately 8–20% of infected individuals with CHB progress to cirrhosis (LC) within 5 years. Furthermore, the 5-year incidence of hepatocellular carcinoma (HCC) in patients with cirrhosis is estimated to be 10% to 17% [1,2,3]. The prevention of complications associated with chronic HBV infection is a major issue worldwide.

Anti-viral therapy with nucleos(t)ide analogues (NUCs) has been proposed to treat chronic HBV infection [4,5]. Recently, studies disclosed that the use of NUC therapy in HBV-related LC patients could reduce the incidence of HCC [6,7]. Nevertheless, the incidence of HCC continues to increase after long-term anti-viral therapy [7,8,9]. Since these NUCs are viral polymerase inhibitors and can only efficiently suppress viral replication [10], the characteristics of the covalently closed circular DNA (cccDNA) means that it cannot be completely eradicated. In the nuclei of infected hepatocytes, it was shown that cccDNA is present, and the amount of cccDNA transcriptional activity in hepatocytes affects the outcome of CHB patients [10].

Hepatitis B core-related antigen (HBcrAg) is a serum biomarker of HBV. HBcrAg is composed of HBeAg, hepatitis B core antigen, and the first 22-kDa of the core protein [11]. Several studies have investigated the clinical use of HBcrAg and the relationships among the serum HBcrAg level, intrahepatic cccDNA levels, HBV DNA levels, and HBsAg level [12,13,14]. The cccDNA level has been used as the intranuclear template for HBV transcripts. Recent studies showed that quantitative HBcrAg could be an alternative marker for intrahepatic cccDNA levels, where higher HBcrAg levels may indicate more active viral replication, contributing to a patient’s HCC risk [13,15]. Previous studies also showed that a high HBcrAg level is independently associated with the development of HCC in CHB-naïve patients [16] or in those undergoing NUC treatment [17,18]. However, most of these studies were cross-sectional and small-scale, and involved the mixed treatment of naïve and experienced non-cirrhotic patients [16,17,18]. The role of HBcrAg in HCC development in HBV-related cirrhotic patients after NUC treatment is unknown.

In the present study, we aimed to investigate the impact of HBcrAg on HCC development in this high-risk cirrhotic population.

## 2. Patients and Methods

This retrospective study was conducted in 1108 treatment-naïve CHB patients with relatively compensated cirrhosis who were receiving long-term oral NUC therapy (including 3TC (30, 2.7%), ADV (7, 0.6%), LDT (93, 8.4%), ETV (728, 65.7%), and TDF (250, 22.6%)). Regular follow-up appointments were conducted in Kaohsiung Chang Gung Memorial Hospital from April 1999 to February 2015. Patients coinfected with hepatitis C or D viruses, and those with decompensated liver cirrhosis with Child’s B or C scores at study entry were excluded from follow-up.

All the patients were tested for HBsAg and HBeAg using commercial assay kits (HBsAg EIA, Abbott, North Chicago, IL; HBeAg EIA, Abbott). Before therapy, none of the patients showed any evidence of HCC. For HBV therapy, the regimens consisted of long-term oral NUC intake. Cirrhosis was diagnosed by using the liver biopsy ex [19] or US findings [20]. US examinations and serum alfa-fetoprotein (AFP) assays were carried out every 3 months. The diagnosis of HCC was carried out using the AASLD guidelines and was based on two typical images (CT-scan and MRI) or was histologically confirmed by liver biopsy [20]. All patients were anonymized and de-identified for data collection. The start date of follow-up was considered as the first day of anti-viral treatment for all patients. The observation period started from the date of the start of anti-viral therapy. The end of follow-up was the date of diagnosis of HCC, the date of death, or the closing date of the study, which was in April 2020.

The HBsAg titers were quantified with Elecsys HBsAg II Quant reagent kits (Roche Diagnostics, Indianapolis, IN), and the HBV DNA content was determined using a hybridization capture kit (HBV test, Roche TaqMan HBV Kit). The serum HBsAg-HQ and HBV core-related antigen (HBcrAg) were measured using a chemiluminescent enzyme immunoassay (CLEIA) Lumipulse G1200 automated analyzer (Fujirebio Inc, Tokyo, Japan). All lab analyses were performed using standard procedures and were accredited by the national lab standard office.

### Statistical Analysis

The Chi-square test was applied to compare the distributions among and between categorical variables. Both t-tests and ANOVA (analysis of variance) tests were used to examine the significant differences between two groups or among three or more groups. The optimal cut-off points for FIB-4 or APRI were derived from the AUROC method, based on all of our subjects. For the concentration of the HBcrAg level, we employed a tertial approach to classify the participants into three groups (33.3%, 66.6%). The time-to-event method was conducted to evaluate the impacts of related baseline biomarkers and characteristics on HCC development in order to estimate the crude hazard ratios (HRs), and adjusted HRs were calculated using univariate and multivariable Cox proportional hazard regression models, respectively. For the multivariable Cox regression models, besides age and gender as compulsive adjusting variables, a stepwise approach was applied for parsimonious Cox regression model selection, where the criterion for a variable being retained or removed was *p* < 0.05. Additionally, the Wald test was applied to examine the significant interaction effects between HBcrAg and other variables. After that, the stratified method was conducted to evaluate the risk of developing HCC if the interaction effect was significant. Based on the multivariable Cox regression results, the directly adjusted survival curve was adopted to generate cumulative incidence curves of HCC development, and the significant confounding factors were adjusted to generate the average cumulative curves. All data analyses were performed using SAS software version 9.4. Results were defined as statistically significant at *p* < 0.05.

## 3. Results

A total of 1108 subjects with chronic B-related liver cirrhosis who were receiving anti-viral treatment were recruited for participation in this study. Regarding the effective treatment outcomes of patients, 219 of 1108 (19.8%) developed HCC within a median follow-up period of 6.8 years. The overall 1-, 3- and 5-year cumulative incidences of HCC were 3.2%, 10.4%, and 15.2%, respectively. Based on the retrospective cohort design, patients with HCC were older (mean age: 64.5 ± 10.5 vs. 60.2 ± 11.5, respectively, *p* < 0.0001), had lower AST (*p* = 0.0135) and ALT levels (*p* = 0.0002), had lower pretreatment platelet counts (*p* = 0.0006), had higher AFP levels (≥20 ng/mL) (*p* = 0.0018), were more likely to have concomitant diabetes mellitus (DM) (*p* = 0.0055), had higher FIB-4 values (≥ 4.1)(*p* < 0.0001), had lower HBsAg-HQ contents (×10^7^ mIU/mL) (≤3 vs. >3, *p* = 0.0086), and had higher HBcrAg contents (>4.9 logU/mL) (*p* = 0.0485). There were no significant differences between the two groups regarding gender, Bil-T, BMI, HBeAg status, HBV DNA levels, baseline creatinine status, or eGFR levels (Table 1).

Among these patients, the tertial concentration levels of HBcrAg were ≤3.4, 3.41–4.9, and >4.9 log U/mL, and the mean ages for patients in these three categories were 63.6, 61.1, and 58.5 y/o, respectively. There was no significant association between the baseline characteristics (gender, platelets, and diabetes) and the HBcrAg level. However, subjects with higher levels of AFP, HBeAg, HBV DNA, and with HBsAg-HQ > 3x10^7^ mIU/mL tended to have significantly higher HBcrAg levels compared with patients with lower levels. However, compared with patients with an FIB-4 level ≥4.1, patients with an FIB-4 level <4.1 had significantly higher levels of HBcrAg (Table 1). Compared with patients with an HBcrAg level ≤3.4, patients with levels of either 3.41–4.9 or >4.9 logU/mL were shown to have a significantly higher risk of developing HCC (Table 2).

Using the time-to-event method to evaluate the impacts of these factors on HCC development, the univariate Cox regression results showed significant crude HRs of 1.74, 1.55, 2.33, 1.51, and 1.72 for platelets < 150, AFP ≥ 20, and FIB-4 ≥ 4.1, having DM and HBsAg-HQ ≤ 3, respectively, compared with their counterpart conditions. For the concentration of HBcrAg levels, compared with patients with values ≤3.4 (lower tertile), the crude HRs were 1.41 (95%CI: 1.00, 1.98) and 1.50 (95%CI: 1.07, 2.10) for patients with HBcrAg concentrations of 3.4–4.9 and >4.9 logU/mL, respectively. Using the multivariable Cox regression model, after adjustment for age, gender, FIB-4, DM, and HBsAg-HQ, the adjusted HRs (adj.HR) were found to be 1.70 (95%CI: 1.21, 2.39) and 2.14 (95%CI: 1.50, 3.05) for patients with levels of 3.4–4.9 and >4.9 logU/mL, respectively. The concentration of HBcrAg was shown to have a significant positive risk impact on HCC development (*p*-value = 0.0001), where higher levels were associated with a greater risk of developing HCC (Table 3). Based on this final parsimonious multivariable Cox regression result, the direct adjusted cumulative curves of HCC incidence were generated for three levels of HBcrAg, and different levels of risk for HCC development are shown (*p*-value < 0.0001) (Figure 1).

A significant interaction effect (*p*-value = 0.0036, not shown in Table 3) between the HBsAg-HQ and HBcrAg concentrations and HCC development was found, which indicates that the effect of the HBcrAg level on HCC incidence is modified by HBsAg-HQ. Therefore, a stratification analysis was conducted to analyze different HBsAg-HQ levels (≤ 3 versus >3). Among subjects with an HBsAg-HQ level >3 × 10^7^ mIU/mL, after adjustment for age, gender, FIB-4, and DM, the HBcrAg level was shown to have an inverse effect on HCC development, but this result did not achieve statistical significance (*p*-value = 0.0706). However, for those with a lower level of HBsAg-HQ (≤ 3 × 10^7^ mIU/mL), compared with those with a level ≤3.4, the adj.HRs were 1.85 (95%CI: 1.30, 2.64) and 2.35 (95%CI: 1.63, 3.40) for patients with levels 3.4–4.9 and >4.9 logU/mL, respectively, indicating that this variable has a significant effect on HCC development (Table 4).

## 4. Discussion

In this study, we evaluated the cumulative incidence of HCC and its associated risk factors in patients with CHB-related LC undergoing NUC treatment. HCC developed in 291 of the 1108 HBV-related LC patients undergoing long-term NUC therapy, with a median follow-up period of 6.8 years (0.6–20 years). The incidence of HCC in our study was higher than that reported in previous studies [16,21]. This discrepancy in HCC development rates might be because all cases in our study had LC, which is associated with a high risk of developing HCC.

In the past, several serologic markers were used as predictors of HCC development in patients with CHB infection. In the present study, we collected novel evidence of the correlation of HBcrAg levels with the clinical and biochemical factors of HBV patients. A strong correlation between HBcrAg levels and HBV DNA was found, suggesting that HBcrAg concentration might be associated virus replication [12]. The HBcrAg level was also found to have a correlation with the HBsAg-HQ level. HBsAg-HQ synthesis is thought to involve cccDNA, as smaller fragments of HBV DNA integrate into the host DNA. HBsAg-HQ is considered a marker of the number of active cccDNA transcripts and the total count of infected hepatocytes [22,23]. We also found a correlation between HBcrAg and the FIB-4 index and APRI, which are noninvasive liver fibrosis markers for CHB patients [24]. Several recent studies have revealed that HBcrAg is a predictor of necro-inflammation and fibrosis in CHB patients, and this has been associated with progress to LC [25,26,27]. In the present study, FIB-4 and APRI were positively associated with the HBcrAg level. Although the mechanism associated with the correlation remains unclear, the results suggest that viruses with a high replication capacity have high cccDNA transcriptional activity. Taken together, the results from our study indicate that HBcrAg may not only reflect virus replication, but also the liver fibrosis status of HBV-related LC patients.

According to previous studies, HBcrAg is associated with the incidence of HCC [28,29]. This has been further validated in untreated patients [16] and patients receiving NUCs [18]. However, most of the patients involved in the previous studies were either already on anti-viral therapy or had a history of HCC. This study provides longitudinal evidence that HBcrAg is an independent HCC risk factor in patients with HBV-related LC. In addition to HBcrAg, HBsAg-HQ levels were also shown to be correlated with the occurrence of HCC in both univariate and multivariate analyses. However, the role of HBsAg-HQ in the prediction of HCC has been more controversial in previous reports [16,18,30].

Serum HBcrAg is a viral marker that is composed of HBeAg, hepatitis B core antigen, and the first 22-kDa of the core protein [11]. As these proteins are proceeded by pre-core/core transcripts, these proteins are not affected by the treatment of NUCs [31]. HBcrAg is considered another suitable candidate for assessing viral replication activity during NA therapy. Furthermore, HBcrAg may be more representative of intrahepatic viral replication than HBsAg, because HBcrAg is unaffected by the pre-core/core transcripts [31]. Previous studies have shown there is a negative correlation between HBsAg and HCC, while there is a positive correlation between HBcrAg and the occurrence of HCC [16,28,29]. The biological and prognostic functions of HBsAg and HBcrAg may differ. Our results further confirm this finding. Notably, the proportion of patients with HCC increased following stratification by increasing HBcrAg titers (Figure 1), but not by HBsAg concentration.

In 2020, Liang et al. associated a higher HBcrAg concentration with a higher risk of developing HCC (hazard ratio = 2.20, >2.9 vs. ≤2.9), based on 1042 CHB HBeAg-negative patients [32]. Similar results were also demonstrated by Kaneko et al. who reported differing risks of the HBcrAg concentration on HCC development between HBeAg-positive/negative patients. A cut-off value of ≥4.1 log U/mL with a hazard ratio of 6.479 (95%CI: 1.334, 34.15) was found based on 139 HBeAg-negative CHB patients (115 pts vs. 24 pts for <4.1 and ≥ 4.1, respectively) [33]. In 2022, Hosaka et al. focused on 180 HBeAg-negative CHB patients to reveal the effect of HBcrAg on the incidence of HCC. Both baseline values and values during anti-viral treatment were shown to play significant roles in HCC development [34]. Recent publications concur that patients with higher HBcrAg levels are at a higher risk of developing HCC, but this was only shown in HBeAg-negative CHB patients. Although a consensus was reached on this result, the cut-off values differed; therefore, more evidence is needed before this marker is used in clinics.

In light of the findings in HBeAg-negative patients, we subjected our data from LC- and HBeAg-negative patients to a multivariable analysis. Compared with patients with HBcrAg levels of ≤3.4, the adjusted HR values were 1.74 (95%CI: 1.22, 2.49) and 1.89 (95%CI: 1.27, 2.79) for patients with levels 3.4–4.9 and >4.9 (not shown in Table 3), respectively. However, no significant results were found among HBeAg-positive patients. This result echoes those found in a previous CHB-based study. Our study is the first to be conducted in liver cirrhosis patients undergoing anti-viral treatment. To determine the cut-off value for HBcrAg that is associated with liver cirrhosis, further studies with large sample sizes are needed. This is also the first study to report that the effect of HBcrAg on HCC is significant among patients with lower HBsAg-HQ levels (≤3 × 10^7^ mIU/mL) and liver cirrhosis, but not in those with an HBsAg-HQ level of >3.

Combining the use of HBsAg and HBcrAg may be a useful way to differentiate the HCC risks levels of HBV-related LC patients. A recent study showed that patients with low HBsAg/high HBcrAg had the highest incidence of HCC [31]. Our large study showed that patients with low HBsAg-HQ/high HBcrAg levels were also at a higher risk of developing HCC in a longitudinal analysis (Table 4). As mentioned earlier, a low HBsAg concentration occurs in the presence of high HBcrAg due to the presence of pre-S/S variants as a result of immune escape. While these pre-S/S variants are associated with HCC development, it may be that HBsAg accumulates in the intracellular endoplasmic reticulum (ER), thus leading to ER stress-dependent and ER stress-independent hepatocarcinogenesis [35,36].

There are several limitations in the present study. First, cirrhosis is only defined by features in ultrasonography, which may not be an accurate enough method. However, the high occurrence rate of HCC might indirectly prove that these patients have advanced liver disease. Second, this study did not test HBV genotypes or core promoter/pre-core mutations because these data are not provided by general clinics and are not covered by National Health Insurance in Taiwan.

In conclusion, as the first study to focus on the assessment of HCC occurrence in the high-risk population of patients with HBV-related LC undergoing NUC treatment, we showed that baseline high HBcrAg and/or low HBsAg levels are independent risk factors for HCC development after NUC treatment. Thus, HBcrAg could be used as a new tumor marker that might surpass the traditional markers of platelets and AFP.

## Figures and Tables

**Figure 1 viruses-14-02671-f001:**
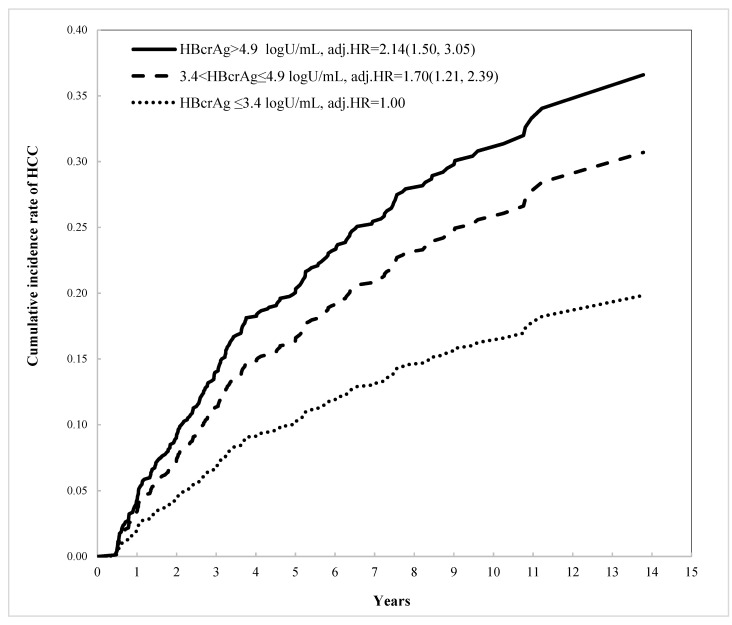
Cumulative incidence rate of HCC by HBcrAg level.

**Table 1 viruses-14-02671-t001:** Characteristics of hepatocellular carcinoma development in chronic hepatitis B patients with liver cirrhosis and complete virological suppression.

Variable		Non-HCC	Incident HCC	*p*-Value
		*n* = 889	*n* = 219	
		No. (%)/Mean (±SD)	No. (%)/Mean (±SD)	
Age (years)	(continuous)	60.2 ± 11.5	64.5 ± 10.5	<0.0001
Gender	Male	661 (79.4%)	172 (20.6%)	0.1990
	Female	228 (82.9%)	47 (17.1%)	
BMI (kg/m^2^)		20.0 ± 4.0	25.0 ± 3.9	0.9921
AST (IU/mL)		156.9 ± 319.0	114.8 ± 194.8	0.0135
ALT (IU/mL)		202.5 ± 451.3	122.0 ± 223.6	0.0002
Bilirubin (mg/dL)		2.6 ± 4.5	3.6 ± 25.8	0.5551
Platelets (×10^9^/L)	<150	373 (85.4%)	64 (14.6%)	0.0006
	≥150	516 (76.9%)	155 (23.1%)	
AFP (ng/mL)	<20	729 (82.1%)	159 (17.9%)	0.0018
	≥20	160 (72.7%)	60 (27.3%)	
HBeAg	-	712 (80.5%)	172 (19.5%)	0.6087
	+	177 (79.0%)	47 (21.0%)	
HBV DNA (IU/mL)	<10^5^	174 (80.9%)	41 (19.1%)	0.7754
	≥10^5^	715 (80.1%)	178 (19.9%)	
Creatinine		2.0 ± 23.7	1.0 ± 0.7	0.1854
EGFR		86.7 ± 28.7	88.4 ± 28.2	0.4288
Diabetes mellitus	No	696 (82.1%)	152 (17.9%)	0.0055
	Yes	193 (74.2%)	67 (25.8%)	
FIB-4	<4.1	374 (87.8%)	52 (12.2%)	<0.0001
	≥4.1	515 (75.5%)	167 (24.5%)	
APRI	<0.9	249 (83.6%)	49 (16.4%)	0.0921
	≥0.9	640 (79.0%)	170 (21.0%)	
HBsAg-HQ (×10^7^ mIU/mL)	≤3	711 (78.7%)	192 (21.3%)	0.0086
	>3	178 (86.8%)	27 (13.2%)	
HBcrAg (logU/mL)	≤3.4	313 (84.4%)	58 (15.6%)	0.0485
	3.41–4.9	288 (78.5%)	79 (21.5%)	
	>4.9	288 (77.8%)	82 (22.2%)	

BMI: body mass index; ALT: alanine aminotransferase; AST: aspartate aminotransferase; CHB: hepatitis B carrier; CHC: hepatitis C carrier; DM: diabetes mellitus; FIB-4: Fibrosis-4 index; APRI: aspartate aminotransferase (AST)-to-platelet ratio index.

**Table 2 viruses-14-02671-t002:** The distribution of HBcrAg levels and other biomarkers.

Variable	Group	HBcrAg (log U/mL)			*p*-Value
		≤3.4	3.41–4.9	>4.9	
		*n* = 371	*n* = 367	*n* = 370	
Age (years)		63.6 ± 11.1	61.1 ± 10.7	58.5 ± 11.8	<0.0001
Gender	Male	272 (32.6%)	273 (32.8%)	288 (34.6%)	0.3303
	Female	99 (36.0%)	94 (34.2%)		
Platelets (×10^9^/L)	<150	140 (32.0%)	142 (32.5%)	155 (35.5%)	0.4799
	≥150	231 (34.4%)	225 (33.5%)	215 (32.1%)	
AFP (ng/mL)	<20	312 (35.1%)	299 (33.7%)	277 (31.2%)	0.0052
	≥20	59 (26.8%)	68 (30.9%)	93 (42.3%)	
HBeAg	-	358 (40.5%)	321 (36.3%)	205 (23.2%)	<0.0001
	+	13 (5.8%)	46 (20.5%)	165 (73.7%)	
HBV DNA (IU/mL)	<10^5^	121 (56.3%)	73 (33.9%)	21 (9.8%)	<0.0001
	≥10^5^	250 (28.0%)	294 (32.9%)	349 (39.1%)	
Diabetes mellitus	No	270 (31.8%)	284 (33.5%)	294 (34.7%)	0.0895
	Yes	101 (38.9%)	83 (31.9%)	76 (29.2%)	
FIB-4	<4.1	111 (29.1%)	146 (34.3%)	169 (39.7%)	<0.0001
	≥4.1	260 (38.1%)	221 (32.4%)	201 (29.5%)	
APRI	<0.9	111 (37.3%)	77 (25.8%)	110 (36.9%)	0.0076
	≥0.9	260 (32.1%)	290 (35.8%)	260 (32.1%)	
HBsAg-HQ	≤3	354 (39.2%)	322 (35.7%)	227 (25.1%)	<0.0001
(×10^7^ mIU/mL)	>3	17 (8.3%)	45 (21.9%)	143 (69.8%)	

**Table 3 viruses-14-02671-t003:** Univariate and multivariable analysis for risk factors associated with the development of hepatocellular carcinoma in patients with a virological response.

Variable	Classification	Univariate		Multivariable	
		HR (95%CI)	*p*-Value	adjHR (95%CI)	*p*-Value
Age(years)	(continuous)	1.03 (1.02, 1.04)	<0.0001	1.02 (1.00, 1.03)	0.0085
Gender	M vs. F	1.24 (0.90, 1.71)	0.1919	1.44 (1.04, 2.00)	0.0298
Platelets (×10^9/^L)	<150 vs. ≥150	1.74 (1.30, 2.33)	0.0002	---	
AFP (ng/mL)	≥20 vs. <20	1.55 (1.15, 2.08)	0.0041	---	
FIB-4	≥4.1 vs. <4.1	2.33 (1.71, 3.18)	<0.0001	2.12 (1.52, 2.97)	<0.0001
DM	Yes. Vs. No	1.51 (1.13, 2.01)	0.0051	1.51 (1.13, 2.03)	0.0052
HBsAg-HQ (×10^7^ mIU/mL)	>3 vs. ≤3	0.58 (0.39, 0.87)	0.0083	0.58 (0.38, 0.89)	0.0131
HBcrAg (logU/mL)	3.4–4.9 vs. ≤3.4	1.41 (1.00, 1.98)	0.0465	1.70 (1.21, 2.39)	0.0001
	>4.9 vs. ≤3.4	1.50 (1.07, 2.10)		2.14 (1.50, 3.05)	

**Table 4 viruses-14-02671-t004:** Multivariable analysis for risk factors associated with the development of hepatocellular carcinoma in patients by HBsAgHQ level *.

Variable	Classification	HBsAg-HQ ≤ 3 (×10^7^ mIU/mL)		HBsAg-HQ > 3 (×10^7^ mIU/mL)	
		adjHR (95%CI)	*p*-Value	adjHR (95%CI)	*p*-Value
Age(years)	(continuous)	1.01 (1.00, 1.03)	0.0459	1.06 (1.02, 1.11)	0.0054
Gender	Male vs. Female	1.54 (1.07, 2.20)	0.0194	0.97 (0.41, 2.29)	0.9411
FIB-4	≥4.1 vs. <4.1	2.26 (1.57, 3.26)	<0.0001	1.11 (0.45, 2.74)	0.8221
DM	Yes. Vs. No	1.55 (1.13, 2.11)	0.0058	1.23 (0.50, 3.05)	0.6568
HBcrAg (logU/mL)	3.4–4.9 vs. ≤3.4	1.85 (1.30, 2.64)	<0.0001	0.22 (0.06, 0.84)	0.0706
	>4.9 vs. ≤3.4	2.35 (1.63, 3.40)		0.40 (0.15, 1.09)	

* The stratified analysis was based on the significant effect of the interaction (*p*-value = 0.0036) between the HBsAg-HQ and HBcrAg concentrations on HCC development.

## Data Availability

Not applicable.

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
