# Peer review of "HBcrAg Predicts Hepatocellular Carcinoma Development in Chronic B Hepatitis Related Liver Cirrhosis Patients Undergoing Long-Term Effective Anti-Viral"

_viruses, 2022, doi:10.3390/v14122671_

Round 1
Reviewer 1 Report (Previous Reviewer 1)
In this study, Chang et al. enrolled 1108 naïve patients diagnosed with HBV-related LC receiving NUC therapy from April 1999 to February 2015. By using a multivariable Cox regression model, they analyzed the relationship between HBcrAg and HCC development in the enrolled patients. The results demonstrated that HBcrAg is an independent risk factor for HCC development in patients with HBV-related LC receiving NUC therapy. Additionally, this correlation was modified by HBsAg-HQ, which leads to the HBcrAg as a predictor for HCC, especially under the condition of HBsAg-HQ levels of <3. The story is intact and interesting, the data is well organized and presented. However, there are some small concerns that need to be addressed.
General comments:
1. In the abstract, what’s the definition of “naïve patients”? The patients here have been treated for several years, does it is qualified for “naïve patients”?
2. In the introduction, the authors should briefly introduce the composition of serum HBcrAg, especially the main components for this parameter (HBcAg, HBeAg, and p22cr...), although it has been clarified in the discussion part.
3. For the abbreviations, when it occurs the first time, authors must show the full name of it. For example, the HBeAg-HQ.
Author Response
"Please see the attachment."

Reviewer 2 Report (Previous Reviewer 2)
Although the authors have made further discussions in the revised manuscript. The weakness of the manuscript was not significantly improved. Without adding new data, the manuscript is not suitable for publication in the current form.
Author Response
"Please see the attachment."

This manuscript is a resubmission of an earlier submission. The following is a list of the peer review reports and author responses from that submission.
Round 1
Reviewer 1 Report
In this study, Chang et al. aimed to investigate the cumulative incidence of HCC and risk factors for the development of HCC in patients with HBV infection-related LC under NUCs therapy. The study enrolled 1108 HBV-related LC patients. By using Chi-square test, Multivariable Cox, and Wald test to analyze the clinical and biochemical factors of these HBV patients, the authors found we showed that HBcrAg was an independent risk factor for HCC development for patients with HBV-related LC after NUCs treatment. Particularly, baseline high HBcrAg and/or low HBsAg levels suggested a significantly higher risk for HCC development. This is an interesting study focusing on the assessment of HCC development in the high-risk population LC-population. However, little new information was provided from this study since there are already several clinical studies that have evaluated the HBcrAg levels for HCC development and English writing, data interpretation in this study needs to be improved largely.
Major comments
1. the English writing very much need to be improved to explain the results clearly and accurately.
2. which anti-viral drugs in this study were used? ETV, TDF, ADV or even 3TC? Different drugs may also affect the viral replication ability, disease progression or even long-term HCC development.
3. Line 123 “Table 2” should be “Table 1” and line125 “Table 1” should be “Table 2”. And where is the data for the conclusion in lines 149-150?
4. The methodology of HBsAg-HQ should be indicated in the methods section.
5. Please read the manuscript carefully to revise the enormous grammatical, word size, and punctuation mark issues.
Author Response
"Please see the attachment."

Reviewer 2 Report
The present study investigated the role of HBcrAg in predicting the hepatocellular carcinoma (HCC) development in chronic hepatitis B related liver cirrhosis receiving long-term anti-viral treatment. The results showed that CHB patients with high HBcAg level were at higher risk of HCC development. There are some major concerns which are listed as below.
1. There have been several studies that demonstrate the predicting value of HBcrAg in HBV-related HCC (Hepatol Commun. 2022;6(1):36-49; J Gastroenterol Hepatol. 2021;36(10):2943-2951; J Gastroenterol. 2020;55(9):899-908.). The authors should highlight novel findings of the present study.
2. Why did the authors use tertial concentration of HBcrAg instead of one cut-off value?
3. What are the genotype of HBV in the enrolled subjects?
4. The written language should be polished by native speakers.
